# Nanoemulsion Stabilized by Safe Surfactin from *Bacillus subtilis* as a Multifunctional, Custom-Designed Smart Delivery System

**DOI:** 10.3390/pharmaceutics12100953

**Published:** 2020-10-10

**Authors:** Agnieszka Lewińska, Marta Domżał-Kędzia, Anna Jaromin, Marcin Łukaszewicz

**Affiliations:** 1Faculty of Chemistry, University of Wroclaw, Joliot-Curie 14, 50-383 Wroclaw, Poland; 2Department of Biotransformation, Faculty of Biotechnology, University of Wroclaw, Joliot-Curie 14a, 50-383 Wroclaw, Poland; marta.domzal@uwr.edu.pl; 3Department of Lipids and Liposomes, Faculty of Biotechnology, University of Wroclaw, Joliot-Curie 14a, 50-383 Wroclaw, Poland; anna.jaromin@uwr.edu.pl

**Keywords:** surfactin, surfactant, *Bacillus subtilis*, delivery system, nanocarrier, skin, stability

## Abstract

The developing field of bio-nanotechnology aims to advance colloidal research via the introduction of multifunctional nanoparticles to augment the dermal effectiveness of active substances. Self-emulsifying drug delivery systems (SEDDS)—isotropic mixtures of oils, surfactants, solvents and co-solvents or surfactants—are attracting interest in the cosmeceutical field. As part of this study, SEDDS systems containing vitamin C or vitamin E and curcumin were developed, whereby the bioavailability of the active compounds increased by enhancing their permeability to deeper layers of the skin. A composition consisting of 50% surfactin from *Bacillus subtilis*, 30% Transcutol and 20% oil phase was designed to encapsulate the active substances, i.e., vitamin C or vitamin E and curcumin, contained in the oil phase. The developed carriers were characterized by average particle sizes of 69–183 nm. The formulations with the vitamins were found to be physically and chemically stable for 6 months. Transdermal tests were carried out, showing that the carriers enable the transport of active substances deep into the skin, stopping at the dermis border. The formulations with vitamin C and vitamin E reduced the discoloration, the vascular lesions, and the depth of the wrinkles on the tested skin, which can be useful in cosmetics in the treatment of problem skin, including capillary and sensitive skin.

## 1. Introduction

Nanotechnology is an expanding strategy that has a significant impact on the development of an efficient delivery system for active substances exhibiting low bioavailability. Concomitantly, alternative methods of administration that do not burden the digestive tract are sought. One of the most promising developments in topical transport is particle size reduction. Such developments, including drug and active substance encapsulation, have attracted much interest, in part due to the advancements in biomaterials enabling the preparation of delivery systems with novel functional properties (size, charge, interfacial functionalization, etc.) through the structural design of nanoemulsions [1].

Nanoemulsions are advanced and promising nanocarriers for the improved delivery of active substances [2]. Compared with conventional emulsions, nanoemulsions provide excellent properties, such as high optical clarity, physical stability and enhanced bioavailability. The small particle sizes of nanoemulsions results in a large surface area, which can be very important for strong interactions. One of the more innovative approaches to active substance delivery is a self-emulsifying drug delivery system (SEDDS). SEDDS are isotropic mixtures of oils, surfactants, solvents and co-solvents or surfactants, forming droplets from 100 to 300 nm in size. They can increase the bioavailability of poorly soluble and permeable compounds, whereby the dissolution step is avoided and permeability through biological membranes is increased owing to the presence of lipids and surfactants [3,4]. Most of the commercialized SEDDS are pharmaceutical preparations, e.g., Sandimmune^®^, Sandimmun Neoral^®^ (cyclosporin A), Norvir^®^ (ritonavir) and Fortovase^®^ (saquinavir).

The uppermost layer of the skin, the stratum corneum, is the main barrier for the penetration of active compounds. This can be overcome to achieve the desired therapeutic and cosmetic effects. Skin penetration enhancement can be achieved either chemically or physically using appropriate formulations [5]. Nanoemulsions can improve topical and transdermal administration by increasing the dissolution capacity of hydrophilic and lipophilic compounds, maintaining a constant flow of drugs from the inner to the outer phase, thereby keeping the outer phase saturation and promoting absorption through the skin [6]. Small droplet sizes result in better adherence to membranes and lead to the more efficient transport of active compounds in a controlled manner [7]. One of the most effective and safest penetration enhancers is water. By hydration of the stratum corneum, the penetration of drugs is increased. The use of moisturizing substances (e.g., urea) to increase the hydration of the stratum corneum leads to an improvement in the diffusion of hydrophilic drugs [5]. Improving the solubility of active compounds and improving its partition coefficient (skin–carrier effect) is another mechanism explaining the action of penetration enhancers [8].

The essential components of a dispersion system responsible for its unique properties are surfactants. The use of surfactants can also improve the penetration of a nanoemulsion through the stratum corneum. Surfactants are amphiphilic molecules that accumulate at the interface between two phases, forming an interfacial molecular film that lowers the solution’s interfacial tension. Increasing awareness of safe and environmentally friendly solutions has encouraged the search for natural-origin surfactants (biosurfactants) with defined structures and advantages [9]. Surfactin (SF), a cyclic lipopeptide produced by *Bacillus subtilis*, is one of the biosurfactants (BS) with unique properties, such as high surface activity, low toxicity, high biodegradability, a broad spectrum of biological activity and ecological acceptability [10]. These favorable features make BS very attractive for many potential applications, e.g., cosmetics, specialty chemicals, foods and pharmaceutics.

The encapsulation of bioactive molecules in natural system mimetics such as nanoformulations increases their bioavailability and biodistribution. This facilitates their controlled release [11], which is particularly important in skin preparations. Antioxidants are the most common ingredients in antiaging cosmetics. A particularly natural choice is the use of vitamin E (Vit E) and vitamin C (Vit C) in cosmetics because of their potent free radical scavenger activity [12,13]. Vit E is an essential nutrient and is considered to minimize the effects of aging owing to its antioxidant and anti-inflammatory properties [14,15]. Tocopherol supplemented on the skin can also help compensate for the loss of endogenous antioxidants observed after UV irradiation [16]. Vit C participates in the metabolism of neurotransmitters, lipids and collagen (a significant natural antioxidant), and catalyzes the oxidation–reduction reactions of cytochrome c [17,18]. It also has a stimulating effect on collagen synthesis [19,20], inhibits tyrosinase, reduces areas with discoloration and provides some UV protection thanks to its antioxidant properties [21,22].

Vit C is relatively stable in the dry state, but in aqueous solutions it decomposes under the influence of such factors as temperature and pH or in the presence of metal ions. Vit E is readily degraded in hostile environments and is sensitive to heat and oxygen, which in turn makes the irreversible conversion to quinone possible [23]. In the form of a nanoemulsion, the encapsulation of Vit C and Vit E can prevent their decomposition and enable their transport through the stratum corneum. It is very important to choose a proper design for the carrier to penetrate the stratum corneum. The stratum corneum is composed of highly keratinized cells and forms a hydrophobic barrier, impeding the transport of active substances. A nanoemulsion can be used to deliver an active substance via transmucosal and transdermal routes, which is why such systems can effectively improve bioavailability [24,25,26].

The key issues in the current surfactant-based formulations are structure–performance relationships and attempts to enhance product biocompatibility with regard to the new biomedical and pharmaceutical nanoscale applications. The main purpose of the present contribution was to explore new nanosized delivery systems. Upon mild agitation followed by dilution in aqueous media, self-emulsifying drug delivery systems (SEDDS) can form fine oil-in-water (o/w) emulsions, microemulsions or nanoemulsions. Thus, these systems can improve the rate and extent of absorption for lipophilic compounds exhibiting dissolution-rate-limited absorption. The present study deals with the design, fabrication and characterization of self-assembly processes and the evaluation of biocompatible nanocarriers and their impacts on the skin. The novelty of this study lies in the application of *B. subtilis*-derived SF as a natural amphiphilic compound that lowers the tension at the interface boundary. The study shows that the encapsulation of ascorbyl tetraisopalmitate and tocopherol in a nanoemulsion based on sunflower oil improves their effectivity and skin penetration. The obtained results will form the basis for the targeted design of natural surfactants to be used in a variety of modern consumer products.

## 2. Materials and Methods

### 2.1. Surfactin Preparation

The preparation of surfactin was based on the method proposed by Jajor et al. [27] with some modifications. Before experimentation, the *B. subtilis* natto KB1 strain was incubated in an agitated (180 rpm) 10 mL LB medium (10 g/L bacto-tryptone, 5 g/L bacto-yeast extract, 10 g/L NaCl) at 37 °C for 24 h. After incubation, the optical density was measured using a microplate reader at 600 nm (Hach ODDYSSEY) to estimate the cell concentration. The strain was grown in modified Landy’s medium (OD_600 nm_ = 0.1) as described before by Jajor et al. [27] with modifications, using 2.3 g/L (NH_4_)NO_3_ instead of ammonia sulfate, 5 mg/L MnSO_4_ and 60 g/L of glucose. Instead of 2 g/L of glutamic acid alone, a mixture of glutamic acid, L-leucine and L-valine (1 g/L of each) was used.

For surfactin production, a 250 mL bacterial culture in a 1 L baffled conical flask was incubated with agitation at 200 rpm and 37 °C for 72 h. After fermentation, the medium was collected and centrifuged (14,000 *g*) at 4 °C for 30 min (Sigma 6K15, rotor 12500, DJB Labcare Ltd., Germany). Then, the bacterial pellets were discarded and the supernatant was acidified with hydrochloric acid to a pH of ~2.0 and left at 4 °C for the next 24 h. The precipitate was collected by centrifugation, 10 mL MiliQ water was added, then the whole solution was neutralized with NaOH. Next, a 5-step liquid–liquid extraction with 200 mL ethyl acetate was carried out. All of the organic phases were collected and evaporated. The residue pellets were dissolved with ultra-pure water and freeze-dried.

### 2.2. Analytical Identity—HPLC and ESI Analysis

The high-performance liquid chromatography (HPLC) system consisted of a Beckmann System Gold 126 pump module equipped with a Knauer variable wavelength monitor and a controller computer with LP-chrom software (Version 1.49, Lipopharm.pl, Gdańsk, Poland, 2017). Chromatography separation was done using a Macherey–Nagel Nucleodur C18 Isis (50 × 4.6 mm, 1.8 microns) column (Fisher Scientific, Göteborg, Sweden).

The mobile phase consisted of acetonitrile and 0.1% aqueous acetic acid, flowing at a rate of 0.7 mL/min in isocratic conditions without temperature control. The solvent ratio was 85:25 for MeCN/aqueous acetic acid. Surfactin detection was performed at 205 nm by means of a UV–Vis Knauer detector. Integration was carried out using the LP-chrom software. The surfactin was analyzed and its concentration was calculated using the sample’s area under the curve (AUC), and the results were compared with the analytical surfactin standard (Sigma-Aldrich, Poznań, Poland). Electrospray ionization (ESI) spectra were obtained using a Bruker apex ultra FT-ICR (ESI-MS) mass spectrometer. The samples were dissolved in dry methanol. The results from the analyses are enclosed in the Appendix A.

### 2.3. SEDDS—Key Components

#### 2.3.1. Co-Solvents

In this study, 2-(2 ethoxyethoxy) ethanol (Transcutol HP; TR) purchased from Gattefossé (Gattefossé SAS, Saint-Priest, France) was used as the co-solvent.

#### 2.3.2. Oils and Active Substances

On the basis of the previous research, Capmul MCM C8 (CA) (Abitec, Janesville, WI, USA) was selected as the oil, which can form nanoemulsions (SEDDS), and as the solvent (matrix) for active substances, creating a double cargo with curcumin (CAC) (Sigma-Aldrich, Poznań, Poland) solubilized in CA.

The active substances used in this study were ascorbyl tetraisopalmitate (trade name Nikkol VC-IP) and tocopherol (trade name Dermofeel toco 70 non GMO). The ascorbyl tetraisopalmitate was a gift from Nicco Chemicals Co. Ltd. (Tokio, Japan). As for the tocopherol, it was a sample received from Evonik Dr. Straetmans GmbH (Hamburg, Germany).

#### 2.3.3. Other Reagents

3-(4,5-dimethylthiazol-2-yl)-2,5-diphenyltetrazolium bromide (MTT), L-glutamine, penicillin and streptomycin were purchased from Sigma-Aldrich (Poznań, Poland). A normal human dermal fibroblast cell line and minimum essential medium Eagle—alpha modifications medium and Dulbecco’s modified Eagle’s medium were purchased from Lonza (Warsaw, Poland). A keratinocyte cell line (CLS Cell Lines Service, Germany), an LB medium, glucose, sodium hydroxide, glutamic acid, L-leucine and L-valine were purchased from BioShop LabEmpire (Rzeszow, Poland). Minerals for modified Landy’s medium were purchased from Chempur (Chempur, Poland). DPPH was purchased from Sigma-Aldrich (Poznań, Poland) and ethanol from POCH (Pol-Aura, Poland). All the other reagents were of analytical grade and used as provided. All the solvents for chromatographic separations were of HPLC grade. The water used in all the experiments was doubly distilled and purified by a Milli-Q water purification system (Millipore, Bedford, MA, USA).

### 2.4. Construction of Diagrams

Phase diagrams were constructed to obtain optimal oil, surfactant and co-surfactant concentrations and component proportions that could result in the maximum micro- or nanoemulsion existence area. A phase diagram in which the triangle vertexes represented the 100% contents of surfactant SF, co-surfactant TR and oil CA was plotted. Various mixtures with varying surfactant, co-surfactant and oil concentrations were prepared. The surfactant, co-surfactant and oil levels varied from 0 to 100% (*w/w*). To prepare the selected composition, the individual components were mixed in the appropriate proportions and then the mixture was subjected to ultrasound while the heating system reached 50 °C. The pre-concentrate prepared in this way was diluted. In order to assess the self-emulsification properties, the formulation was introduced into 10 mL to 1 L of water in a glass Erlenmeyer flask at 37 °C. Mixing was done in a beaker using a stirring bar and a magnetic stirring plate. After mixing with water, the phases were categorized on the basis of visual observation. The tendency to form a transparent emulsion was judged as good, whereas poor or no emulsion formation was considered bad [28]. The prepared formulations were evaluated using DLS analysis and imaging by TEM.

### 2.5. Characterization of SEDDS System Stabilized by Biosurfactant

#### 2.5.1. DLS Analysis

The average particle size (D_H_), polydispersity index (PdI) and zeta potential (ξ) values of the droplets were determined by dynamic light scattering (DLS). The measurements were performed using a Zetasizer Nano ZS (Malvern Instruments, Malvern, UK) with a detection angle of 173°, equipped with a He-Ne laser (632.8 nm) and an ALV 5000 multibit, multitau autocorrelator (Malvern Instruments, Malvern, UK). All of the DLS measurements were performed at 298 K. The DTS (nano) program was used to evaluate the data. Each value was calculated as the average of three subsequent instrument runs with at least 20 measurements.

#### 2.5.2. TEM Microscopy

Transmission electron microscopy measurements were performed using an FEI Tecnai G2 20 XTWIN transmission electron microscope (FEI, Hillsboro, OR, USA). The size distribution for each nanoemulsion sample was determined by counting the sizes of approximately 250 nanoparticles from several TEM images obtained from different places of the grid. Energy dispersive X-ray (EDX) spectra were recorded using a Thermo Scientific Ultra Dry detector (Thermo Fisher Scientific, Waltham, MA USA) (resolution 129 eV) and analyzed using a Noran System 7 (Thermo Fisher Scientific, Waltham, MA USA). A few drops of the diluted suspension were placed on the Cu-Ni grid and stained with 2% uranyl acetate before capturing the image. The size distribution plots were fitted using a Gauss curve approximation.

#### 2.5.3. Stability Studies

The nanoemulsions were subjected to time-dependent size (D_H_) and ξ potential measurements at room temperature (RT) as the dispersion stability test. The nanoemulsions that did not show creaming, sedimentation, coalescence or flocculation during the tests were selected for turbidity testing. The measurements were performed for all the freshly prepared nanoemulsions and after 30, 90 and 180 days.

#### 2.5.4. Scavenging Free Radicals

Aliquots of the stable radical solution in ethanol were stirred in the dark for 2 h to obtain a homogeneous solution. Then, samples were prepared, containing 25 μL of the radical (DPPH) and 35 μL of the nanoemulsion sample. A capillary tube was sealed and put inside a standard EPR quartz tube placed in a resonant cavity. The procedure was carried out at room temperature. The data were reported as the averages of three measurements. The EPR spectra were recorded using a Bruker Elexsys 500 spectrometer operating in the X-band frequency (~9.7 GHz). A microwave power of 4 mW, a modulation amplitude of 1 G, a sweep width of 100 G, a time constant of 40 ms and a conversion time of 160 ms were adopted. An analysis of the EPR spectra was carried out using the WinEPR software package, version 1.26b (Bruker WinEPR GmbH, Rheinstetten, Germany). The double integral of the signal was evaluated as representative of the free radical concentration. The field under the absorption curve was proportional to the amount of stable radicals remaining in the sample.

### 2.6. Biological Evaluation

#### 2.6.1. Cytotoxicity Assay

The normal human dermal fibroblast cell line (NHDF) was cultured in the minimum essential medium Eagle (EMEM)—alpha modification medium supplemented with 10% fetal bovine serum, 2 mM glutamine, 100 U/mL penicillin and 100 μg/mL streptomycin. The HaCaT cell line was cultured in Dulbecco’s modified Eagle’s medium (DMEM) with 10% FBS, 2 mM glutamine, 100 U/mL penicillin and 100 μg/mL streptomycin. The cells were seeded in triplicate into 96-well culture plates at 6 × 10^3^ and 4 × 10^3^ cells/well for NHDF and HaCaT cells, respectively. The seeded plates were cultured in a 37 °C incubator in a humidified atmosphere containing 5% CO_2_. The cells in triplicate wells were treated with an appropriate volume of specific nanoemulsions to obtain 62.5, 125, 175 and 250 µg/mL, then incubated for 24 and 48 h. Cell viability was determined by the MTT method. MTT solution (50 μL) was added to each well and incubated at 37 °C for 4 h. After the incubation, the medium was removed and 50 µL of DMSO was added to dissolve the formazan crystals. The absorbance was measured at 570 nm on an ASYS UWM 340 microplate reader (Biogenet, Józefów, Poland). The untreated control was normalized to 100% for each assay and the treatments were expressed as percentages of the control. The cell viability was calculated as:$$\mathrm{Cell}\;\mathrm{viability}\;\left(\%\right)=\;\frac{{\mathrm{Absorbance}}_{570\;\mathrm{nm}\;\left(\mathrm{sample}\right)}}{{\mathrm{Absorbance}}_{570\;\mathrm{nm}\;\left(\mathrm{control}\right)}}\times 100\%$$

The assays were performed in triplicate and the data were expressed as mean values ± standard deviations.

#### 2.6.2. Ex Vivo Skin Transdermal Studies

Experiments were performed as described by Lewińska et al. [1] and Jaromin et al. [29]. Full-thickness pig skins were cleaned with tap water, separated from the underlying cartilage, then the subcutaneous fat was cut and stored at −20 °C for up to 1 month after preparation to ensure skin integrity. Before use, skin samples were thawed to room temperature and hydrated with 0.05 M phosphate buffer (pH 7.4) for 30 min. Formulations with an SF/TR/CA *w/w* composition of 50:30:20 together with coumarin, which was previously dissolved to saturation in the oil phase, were used in this experiment. Then, 20 mg of the prepared oil phase was added to the formulation. In the case of coumarin, its saturated solution in CA was prepared and then the formulations were prepared as described above. As the control, coumarin dissolved to saturation in ascorbyl tetraisopalmitate was used. In the experiment, the skin was cut to size and was accurately mounted and clamped between the donor and the receptor compartments of a set of transdermal Franz diffusion cells (Hanson Research Corporation, Chatsworth, CA, USA). The system was equipped with a heating circuit with vertical glass diffusion cells with a jacket. The receptor cell was filled with 0.05 M phosphate buffer at pH 7.4 and gently stirred with a magnetic stirrer at 350 rpm. Then, 0.5 mL of each SEDDS system was added into the donor compartment on the stratum corneum side of the skin. The system was adjusted to 32.5 °C. The study was conducted for 4 h and then the treated skin samples were removed from the Franz diffusion cells. The fluid taken from the Franz acceptor chamber was subjected to HPLC analysis (for SF) and UV–Vis analysis (for active cargo) to identify the active compounds in the formulations.

#### 2.6.3. Visualization of Penetration

Confocal laser scanning microscopy (CLMS) containing coumarin as a fluorescent marker was used to visualize the penetration of the preparation and to determine the place it reached. The skin was fixed in 4% (*w/v*) paraformaldehyde in phosphate-buffered saline (PBS) at room temperature for 45 min. After incubation, the samples were washed 3 times for 5 min in PBS, transferred to a 30% sucrose solution (*w/v*) in PBS and stored at 4 °C overnight. The next day the samples were embedded in a cryopreservation solution (Sakura, Torrance, CA, USA), placed in a cryomold and frozen. The samples were cut into sections in a cryostat at −20 °C and placed on Superfrost Plus microscope slides (Sigma-Aldrich, St. Louis, MO, USA). An Olympus Fluo View FV1000 confocal laser scanning microscope (Olympus, Tokyo, Japan) was used for imaging. The images were processed using the FV10-ASW_Viewer and ImageJ (3.0, Olympus America Inc., Center Valley, CA, USA, 2011; 3.0, Bioz, Inc., Los Altos, CA, USA) [30].

#### 2.6.4. In Vivo Evaluation of Nanoemulsions—Skin Interaction Study

A NatiV3 analyzer (Beauty of Science, Wrocław, Poland) was used for in vivo evaluation of volunteers (age 30–50). The analyzer is a comprehensive device for computer cosmetology diagnostics. It uses a physical measurement system and a modern digital camera working with HD-ready technology. The 2-in-1 measuring system makes physical and optical analyses of the skin possible, whereby the condition of the skin can be objectively assessed. It has a built-in camera with 40× magnification. Four types of light are used: cold white 3000K, white normal 6000K, warm white 800K and UV 405 nm. NatiV3 enables the assessment of skin parameters, such as skin structure, oiling, peeling, pore size, wrinkle size, vascular lesions, discoloration and hydration. The measurements were analyzed using the manufacturer’s database. The software and the parameters had been developed on the basis of more than 2500 studies of people aged 18–90 years, both women and men. The prepared 1 mL nanoformulations were applied on a specific area of the facial skin daily. In order to precisely define the place of application, a prepared matrix with 1 × 1 cm squares was cut out. The measurements by means of the analyzer were carried out before the application of the formulations and 14 and 28 days after their application. The skin parameters, namely wrinkle size, vascular lesions and discoloration, were evaluated. The study protocol was approved by the Bioethics Commission at the Lower Silesian Chamber of Physicians and Dentists (1/PNHAB/2020).

### 2.7. Statistical Analysis

All the data are expressed as the mean values ± standard deviation of three measurements. Statistical analyses were performed using the one-way analysis of variance with the post hoc Tukey honest significant difference calculator (GraphPad Prism). A value of *p* < 0.05 was considered to be statistically significant.

## 3. Results and Discussion

Supplying active substances to deeper layers of the skin is very challenging. Most cosmetic preparations do not penetrate through the stratum corneum to deeper layers of the skin. It is assumed that carriers of active substances should be composed of biodegradable components that undergo enzymatic degradation to the compounds naturally occurring in the human body. According to the published results, carriers exceeding 40 nm are not able to penetrate the living cells of the epidermis or pass into the bloodstream, and so cause no adverse effects [31]. The use of an active substance, and especially its uncontrolled release, can lead to a hypersensitivity reaction. An extremely important factor determining safety is the proper choice of surfactants, which means that only compounds that have been approved by the American Food and Drug Administration (FDA) and are on the Generally Recognized as Safe (GRAS) list can be used in the preparation of carriers. Besides particle size and biodegradability, attention should be paid to the route of administration. The design of active substance carriers was intended to enable the delivery of hydrophobic substances to the deeper layers of the skin. The high lipophilicity of active compounds is a major obstacle to their use. However, this problem has been solved by introducing active substances into the core of a hydrophobic nanoemulsion system with a modified hydrophilic surface. In this way, it is possible to dermally transport a lipophilic substance and then release it.

### 3.1. Optimization, Characterization and Stability of Smart SEDDS Systems Obtained through Nanoemulsion Structural Design

The combination of biomaterials—especially biosurfactants and active substances—enables the development of intelligent formulations and opens up enormous possibilities for biomedical applications. The production of optimal SEDDS requires relatively high concentrations (generally over 30% *w/w*) of surfactants. Co-solvents, such as ethanol, propylene glycol and polyethylene glycol, can act as co-surfactants in emulsion systems. Solvents can help to dissolve large amounts of hydrophilic surfactants or a hydrophobic drug in the lipid base. Alcohols and other volatile solvents evaporate, which can lead to the precipitation of the active substances encapsulated in SEDDS. The nanoemulsions obtained in this study were stabilized with a natural biosurfactant, i.e., a sodium salt of SF, and prepared using the SEDDS technique. SF is a natural biosurfactant characterized by high surface activity, low toxicity and high biodegradability, whereby it is highly suitable for use in epidermal formulations. Oil is one of the most important substances in creating SEDDS. It enables the dissolution of lipophilic active compounds, facilitates self-emulsification and increases the transport of active substances to the deeper layers of the skin. Capmul—a mixture of mono and diglycerides of medium chain (mainly caprylic) fatty acids—was used as the model oil phase for the construction of phase diagrams. It is an excellent solvent for many organic compounds and a useful emulsifier for water–oil (w/o) systems. Pure surfactants often organize themselves well at the liquid–liquid boundary, which results in relatively stiff interfaces and even liquid–crystal phases. Co-surfactant Transcutol HP was used to achieve ultralow interfacial tension. It was added to the process to enhance the effectiveness and the oil-solubilizing capacity of the nanoemulsion systems. Phase diagrams were constructed with various concentrations of the selected oil, SF and the co-surfactant (the SF, co-surfactant and oil phase weight ratios of 0.01–96.99:0.01–96.99:3–70% *w/w* were tested). As nanoemulsions form upon dilution in aqueous media, they are often described as microemulsion pre-concentrates. A ternary phase diagram was constructed and used to identify the region of efficient self-emulsification. By constructing phase diagrams, one can study the phase behavior of surfactant systems. In this way, one can obtain information on the different phase boundaries as a function of the composition variables, and more importantly one can choose the compositions, which after spontaneous dilution (50 mg of a given composition is added to 10 mL of distilled water) form o/w nanoemulsions. This method represents an effective approach to nanoemulsion preparation. It is a low-energy emulsification technique, making use of the chemical energy stored in the components. As a consequence, low-energy emulsifications offer advantages in terms of their low cost, higher energy efficiency and simplicity of implementation [32,33]. The choice of an appropriate surfactant is extremely important for controlling the functional properties of nanoemulsions, including their long-term colloidal stability and further interactions with biological systems. The surfactant is adsorbed at the oil–water interface, reducing the interfacial energy, as well as providing a mechanical barrier to coalescence or other nanoemulsion destabilization processes. As a model for further research, a phase triangle with CA as the oil phase was developed. Antioxidant vitamins, including vitamins C and E, play important roles in the cosmetics industry. The antioxidant properties of vitamin C help to delay the aging of skin cells and the sealing of blood vessels. Vitamin C also has anti-inflammatory and discoloration brightening properties. Vitamin E is known as the “vitamin of youth” because it inhibits the aging process of the body. Vitamin E also has valuable nutritional, moisturizing, oiling and regeneration properties. Since these compounds are some of the strongest antioxidants, it was decided to use them as the oil phase in the prepared formulations. CA had already been used in the development of SEDDS systems for oral drug delivery and it might enhance the solubility of poorly water-soluble substances [34,35]. In the present study, optimized SF, TR, CA, vitamin C and vitamin E contents were selected, considering the self-emulsification properties of the formulations upon their addition to water under mild agitation conditions. Visual observations were carried out for transparent and easily flowable oil-in-water (o/w) nanoemulsions after dilution. Favorable weight ratios were selected for the individual components on the basis of the developed phase triangle (S3). The amounts were 10–60:10–90:10–65 *w/w* for the biosurfactant, the co-surfactant and the oil phase, respectively. The strategy used for the assembly of the self-emulsifying drug delivery systems formulated by mixing surface, matrix and cargo components is presented in Scheme 1.

Considering that this type of surfactant-stabilized carrier was new and had not been described in the literature before, we decided to carry out studies to determine the impact of its composition and subsequent dilutions on the physicochemical properties of the formulations. For this purpose, we selected three model compositions stabilized with increasing amounts of SF, namely 10, 30 and 50%, tested both at 100-fold and 1000-fold dilutions. The selected composition samples were characterized by their droplets size (D_H_), polydispersity (PdI) and ζ-potential. As can be seen in Figure 1A, the presence of both 30 and 50% SF in the composition leads to a drastic reduction in the size of the obtained nanoemulsions at the 100-fold dilution. At the 1000-fold dilution, this effect is also visible, although the difference is smaller. It is worth noting that the composition with 10 SF and the one with 30% SF contained the same amount of oil phase (50%). This phenomenon suggests that the biosurfactant plays a key role in stabilizing this type of structure, and is indicative of the excellent compatibility of all of the components in the amounts used. As expected, a higher dilution resulted in a proportional decrease in the next considered parameter, namely PdI (Figure 1B). The best parameters, expressed by the lowest PdI value, are for compositions containing 50% SF, which indicates their excellent homogeneity. On the other hand, the increasing SF content lowers the zeta potential, which is further reduced as a result of further dilution (Figure 1C). Nevertheless, the obtained values are very high (e.g., in the range of −75.9–−92.6 mV for the 100-fold dilution), which may suggest their high stability. The size distributions of the nanoemulsions with vitamin C and vitamin E after preparation and after 195 days are shown in Appendix A. The nanoemulsion containing curcumin dissolved in CA was characterized by a small particle size. However, as can be seen in Table 1, the PdI of this formulation is higher than the Pdl values of the other formulations. Additionally, aggregation of molecules was observed in the nanoemulsions. Smaller particle sizes ranging from 20 to 50 nm were obtained by Yousef et al. for a nanoemulsion with curcumin [36]. In their study they used other surfactants, which suggests that the choice of surfactant may influence the size of the particles.

On the basis of the above results, for further research we chose the 50:30:20 (SF/TR/oil *w/w* %) composition, proving its universality by using a different oil phase (Appendix A) and a different dilution. Table 1 presents data describing o/w nanoemulsions, containing vitamin C or E and CA as the oil phase at a 200-fold dilution. It can be seen that the use of these oils resulted in an increase in the size of the obtained nanostructures and an increase in PdI. Additionally, the zeta potential increased, which is a very positive phenomenon. In summary, it can be stated that despite the larger particle sizes of the nanoemulsions, they are still within the acceptable size range for skin applications and are characterized by high zeta potential values, indicating high stability, which is very important during the production process.

Transmission electron microscopy and scanning electron microscopy were used to determine the morphology of the obtained nanocarriers. Representative images obtained for the systems containing CA, vitamin C or vitamin E showed the presence of well-separated (practically with no aggregation) spherical droplets, with a relatively narrow size distribution (Figure 2). The obtained results indicate that the droplets are homogenous and sufficiently stabilized. The morphology of the unloaded nanoemulsion system (Figure 2A) indicates a more spherical carrier shape than in the case of the system loaded with the additional active substance (Figure 2B). As for the carrier with encapsulated curcumin, its shape differs from that of the other tested formulations. After loading curcumin into the CA and preparing the nanoemulsion, it was observed that the droplets lost their spherical shape. The morphology of the CAC nanoemulsions (Figure 2B) was characterized by a more spindle-like shape, and the spherical structures became flattened and elongated. This feature and the specific zeta potential may suggest instability of the prepared nanoemulsion. Representative transmission microscopy images are shown in Figure 2.

The stability of a nanoemulsion is one of the most important factors for any potential biological application. Such nanostructures when not stabilized electrostatically are usually metastable due to the short-range van der Waals attraction. In order to avoid aggregation due to their low colloidal stability, steric or electrostatic repulsion can be applied for stabilization. The nanoemulsions with the 50:30:20 composition (SF/TR/oil *w/w* %, where oil is either CA, Vit C, Vit E) were first visually assessed and no changes in the physical appearance of the formulations were observed. After 7 days, they remained clear with no signs of creaming, sedimentation, flocculation or coalescence. The samples were stored at 25 °C for 195 days and then their stability was evaluated based on the particle size (D_H_), particle size distribution and zeta potential. The literature data indicate that sufficiently high and sufficiently low potential zeta values (up to 30mV and below −30mV, respectively) enhance the stability and functionality [37,38,39]. Unfortunately, the storage period had a destructive effect on the systems with solubilized curcumin in CA—flocculation was observed, while systems with Vit C and Vit E remained clear, with slight opacity. The hydrodynamic diameter of the nanoemulsion with Vit C slightly increased, reaching 190.57 ± 0.3 after 195 days. In the case of the system with Vit E, a decrease in diameter down to 147.6 ± 1.9 was observed. Nevertheless, both the decrease of 19.74% for Vit E and the increase of 7.99% for Vit C resulted in system stabilization. Both formulations were characterized by good PdI values of 0.183 ± 0.06 and 0.126 ± 0.016 for Vit E and Vit C, respectively. The zeta potentials (−77.57 ± 0.8 and −89.7 ± 1.14) suggested that the systems had good stability. The formulations containing the vitamins were found to be physically and chemically stable for about 195 days at the room temperature of 25 °C.

### 3.2. Biological Evaluation

#### 3.2.1. Cytotoxicity Assay

Any proposed delivery system must be compatible with the physiological environment. As was proven previously, the nanocarrier’s charge and composition have a bearing on the key physicochemical issues affecting the ability of nanoparticles to interact with cells. The nanoemulsions with Vit C or Vit E and curcumin as the active substances and CA as the model oil phase were used as the starting compositions in the testing (using the MTT assay) of skin cell cytotoxicity prior to dermal exposure. The results of the viability of normal human dermal fibroblasts (NHDF) and keratinocytes (HaCaT) treated with different concentrations of the formulations for 24 and 48 h are presented in Figure 3. In the case of the NHDF cell line, concentrations in the range of 62.5–175 µg/mL resulted in viability values >70% after 24 and 48 h. For all the tested concentrations, the formulation with Vit E showed no toxic effect against NHDF with a viability of over 70% after 24 and 48 h. The formulations tested against the keratinocytes (HaCaT) showed no cytotoxic effect for only one concentration (62.5 µg/mL) when treated for 24 and 48 h.

It is quite difficult to determine the cytotoxicity of multicomponent formulations because of the possible specific interactions between the components. Hence, it was decided to compare the obtained results with the documented cytotoxicity of the individual compounds used in the nanoformulations. The nanoformulations were tested after their dilution, which means that the amounts of the particular nanoemulsion components were in the following ranges: 120–30 µM SF, 560–10 µM TR, 370–60 µM CA, 180–30 µM Vit C, 170–30 µM Vit E, 200–30 µM CAC. The compositions were as described before: 50:30:20 for SF/TR/oil *w/w* %, where the oil phase was CA, Vit C, Vit E or CAC.

The determined IC_50_ values of surfactin C-15 against the HeLa cell line were 86.9, 73.1 and 50.2 μM after 16, 24, and 48 h, respectively [40]. Human colon cancer cells (LoVo) treated with 80 µM surfactin for 24 and 48 h at almost 80% underwent death and 30 µM surfactin applied for 24 h also displayed significant antiproliferative activity [41]. For normal cells, the 50% cytotoxic concentration of antimicrobial lipopeptides (surfactin and fengycin) for porcine kidney (PK-15) cells amounted to 32.87μmol/L, while that for HaCaT cells amounted to 100 μg/mL, the values for which were higher than those for other tested tumor cells [42,43].

The vitamin C used in the research occurs in various forms, such as ascorbic acid or ascorbyl palmitate. Ascorbic acid in concentrations above 10 mM was found to be strongly toxic to human tenocytes and at concentrations from 1 mM adversely affected cell morphology [44]. Other research showed that the cytotoxicity of vitamin C in the form of L-ascorbic acid and ascorbyl palmitate to L929 fibroblasts was in the range of 0.000015–0.15% [45]. The IC_50_ values for vitamin C tested against cell lines CLV102, ME18 and ME18/R amounted to 200 ± 13.36, 373 ± 11.70 and 310 ± 10, respectively [46]. As for vitamin E, it is an antioxidant nutrient. The vitamin E dose ranges used in clinical practice vary widely. Generally, the dose of 5–80 µM vitamin E is used in clinical practice in combination treatment for various diseases [47]. The IC_50_ values for vitamin E tested against cell lines CLV102, ME18 and ME18/R amounted to 500 ± 31.18, 500 ± 26.31 and 500 ± 32.99, respectively [46].

TR dose-dependently inhibits the proliferation of 3T3 murine, normal and psoriatic human fibroblasts. The proliferation of all the lines was reduced when tested against 2, 1 and 0.5 mM concentrations of TR. In the case of human psoriatic dermal fibroblasts, 0.5 mM TR still had an antiproliferative effect, while for human normal dermal fibroblasts 0.1 mM and 0.01 mM concentrations were poorly effective [48]. In the study conducted by Alvi and Chaterjee, TR showed an IC_50_ value of 2.0 ± 0.35% (*v/v*) when tested against Caco-2, while Labrasol (another non-ionic surfactant) was much more toxic (IC_50_ 0.22 ± 0.01%) to this line [49].

Numerous studies on curcumin indicate its anti-inflammatory, antioxidant and anticancer effects. After 24 h, curcumin reached an IC_50_ < 200 μM against the NHDF cell line [50]. Some reports indicate that pure curcumin concentration for oncogene target inhibition and for the inhibitory effect on cancer cell proliferation is ∼15–20 μM [51].

Capmul is a substance used for various purposes in research on various types of formulations. However, in studies on skin formulations, Capmul was rejected during carrier optimization [52] or its effect on human skin cells was not determined [53]. Capmul^®^ MCM at concentrations ≥ 0.625% (*w/v*) damaged the monolayer of the Caco-2 cell line in studies by Keemink and Bergstrom [54]. However, in the research by Roohinejad et al., a microemulsion (76.6 ± 3.1 nm) containing 0.038% Capmul^®^ MCM and 0.025% Tween 80 reduced the survival of the same cell line by 40%, but this effect was only due to the Tween80 being used [55]. In general, Capmul^®^ MCM is considered safe and has the Generally Recognized as Safe status (GRAS) (21 CFR § 184.1505).

Any carrier that enables the delivery of active substances to tissues must be compatible with the physiological environment. The safety of nanoemulsions is increasingly being studied. Their effect on human cells depends not only on the type of cells, but also on the ingredients used and their interaction with each other. The obtained results suggest that the size of a nanoemulsion does not influence its cytotoxicity, which is in agreement with the literature data [1].

#### 3.2.2. Scavenging of Free Radicals

The 2,2-diphenyl-1-picrylhydrazyl (DPPH•) shown in Figure 4G was used to measure radical scavenging activity (RSA). The method is based on the ability of the antioxidant to scavenge DPPH•. When reacting with a substance, a stable free radical with an unpaired electron produces changes that can be monitored by EPR spectroscopy. The method is widely used in antiradical assays, as it is thought to be one of the most reliable methods. It has several advantages over other electromagnetic radiation spectroscopy methods, i.e., it is free-radical-specific (no other substances appear in the registered signal); the energy used for the microwave irradiation of the sample is insignificantly small, whereby the possibility of damage to the analyzed sample is negligible; and a small sample is needed for the test. Moreover, the DPPH• has a well-defined EPR spectrum and it is very easy to monitor the decreasing signal intensity after the addition of antioxidant compounds. The field under the absorption curve is proportional to the amount of stable radicals remaining in the sample. The relative concentration of DPPH• was calculated from the calibration curve obtained from the dependence of DI on C (mol/dm^3^), where DI is a value obtained from the double integration of the EPR spectra of the standard solutions. Regions of integration were selected to cover the resolved signals of the hyperfine structure (Figure 4F).

An analysis of the EPR spectra results presented in Figure 4 indicated a non-linear relationship between the DPPH• concentration and time. The record of the DPPH loss revealed a marked difference in the reactions for the different nanoemulsions. An immediate very fast reaction (a rapid decrease in the remaining DPPH) was observed for the systems with vitamin C and vitamin E, whereas the nanoemulsions with CA and CAC showed much slower reaction rates. The surface of the area under the signal curve depends on the power (kind) of the antioxidant and diminishes over time. As noted previously, antioxidants can quench DPPH by both electron and hydrogen atom transfer. The electron transfer is very fast and not diffusion-controlled, whereas the hydrogen atom transfer is slower and can be controlled by diffusion [56]. Hence it can be concluded that the compounds that reduce DPPH quickly act only through electron transfer. Compounds that reduce DPPH more slowly and require an extended time for the reaction to be completed act through hydrogen atom transfer, steric hindrance or both. Therefore, one can conclude that in the considered case DPPH reduction takes place via hydrogen atom transfer. The most effective scavenging of free radicals in the first few minutes was demonstrated for nanoemulsions with Vit E and Vit C, which resulted in 2.7% and 18% of free radicals remaining after 5 min, respectively. For nanoemulsions with CA and CAC, the scavenging was very weak (almost unnoticeable for CA). It can be concluded that the nanoemulsions with vitamin C and vitamin E exhibit antiradical properties.

#### 3.2.3. Ex Vivo—Transdermal Studies

The stratum corneum consists of a series of layers of specialized skin cells that are continuously shedding, the role of which is to protect the inner layers of the skin [57]. Most of the chemical substances topically applied to the skin are not able to penetrate the stratum corneum [58]. Skin penetration enhancement can be achieved chemically, physically or through the use of appropriate formulations [5], hence leading to the conclusion that nanoemulsions more effectively deliver active ingredients deep into the skin, making the formulation more effective. By transporting a nanoemulsion through the stratum corneum to the deeper layers of the skin, one can achieve better performance of the encapsulated active substances. The goal of this research was to confirm the active topical permeation of the nanocarrier. Diffusion studies were carried out in a Franz chamber. The tested system was placed in the donor chamber, and after 1 h the solution was collected from the acceptor chamber. The solutions were subjected to UV–Vis analysis (for oils and active substances) and HPLC analysis (for SF) according to the classical methods. The considered compounds were not identified in any of the tested formulations. During the permeation study in the Franz chamber, the entire formulation remained on the membrane (skin), which indicates that the tested dispersion system did not penetrate to the bloodstream. In order to determine the location of the studied nanoemulsion, the skin from the Franz chamber test was subjected to confocal microscopy imaging (Figure 5).

It can be concluded from the analyses that coumarin suspended in oil is not able to penetrate deep into the skin (Figure 5A,B). The prepared formulation forms an occlusive layer on the surface of the stratum corneum. However, one can see that thanks to coumarin encapsulation in the nanoemulsion, the latter penetrates the skin (examined 1 h after application), stopping at the dermis border (Figure 5C,D). Similar results were obtained by Sarheed et al. [59], Yousef et al. [36] and Nastiti et al. [60], whereby the encapsulation of loratadine, curcumin and resveratrol, respectively, in a nanoemulsion enhanced the permeability of these compounds through the skin.

#### 3.2.4. In Vivo Evaluation of Nanoemulsions—Skin Interaction Study

As part of this study, the skin parameters (wrinkle size, vascular lesions and discoloration) measured and evaluated with the NatiV3 analyzer after the application of the nanoemulsions with vitamin C and vitamin E (Table 2). Vitamin C is a compound known for its antipigmentary properties. Vitamin E is used to minimize the effects of aging. It has well-documented antioxidant and anti-inflammatory properties, whereby it protects the skin against oxidative stress. The optimal concentrations of the active substances depend on the formulation and on the delivery method, although skin changes are difficult to follow and become visible over time. The results reported here come from a statistical analysis based on the algorithm developed by the manufacturer. First, the structure of the examined skin was rated using the database developed by Beauty of Science [61] and the image of the skin’s surface evenness was examined. A skin structure value defines the average smoothness of the lines, which determines the surface of the skin in the test area. The skin structure (SE) is defined as correct, disturbed or incorrect when the parameters are in the ranges of 0–10.99, 11–12.99 and 13–16, respectively. On the basis of the measurements made using the NatiV3 analyzer, the studied skin structure was identified as correct (Figure 6).

The system used in our study is able to deliver active compounds to the deeper layers of the skin. Hence, much lower concentrations of the active substances to be delivered can be used. The analyses showed a reduction in discoloration and in vascular lesions after 28 days of testing for the formulation containing Vit C in comparison with the control formulation (Figure 7). A similar study of the effect of a topical formulation with 25% ascorbic acid and a chemical penetration enhancer showed a significant decrease in pigmentation caused by melasma after 16 weeks of use [62]. The present ex vivo study was conducted for 4 weeks, which may be too short a period of time for the effects of the therapy to be become more visible, in comparison with the study by Hwang et al. Moreover, in [62] there is no information on the condition of the skin after 28 days of using the preparation. Vit C enhances the endothelial synthesis and deposition of collagen type IV in the formation of the basement membranes of blood vessels [63], reduces their vascular fragility [64] and seals them [65]. The most important issue is the optimal cellular content of ascorbate. It appears that low millimolar ascorbate concentrations are normal in most animal tissues, in human leukocytes and probably in the endothelium [63]. In the present study, changes in vascular lesions were observed. Even though the tested skin was quite healthy, some vascular lesions were visible on the skin. These were reduced by 30.71–64% and 85% after 14 and 28 days, respectively. In comparison, in the research conducted by Jaros et al., a 5% Vit C concentrate was effective in treating capillary and photograph-aging skin. The above authors obtained 9 and 16% reductions in erythema after 2 and 4 weeks, respectively [65].

Topical α-tocopherol is mostly used at a concentration of 5% or less [66]. The highest a-tocopherol levels were found in the lower stratum corneum, whereas the lowest levels were present in the upper layers [14]. There are no published data on dose-response studies concerning the optimal dosage of Vit E. At concentrations of less than 0.2% in rinse-off products, α-tocopherol leads to increased levels of Vit E in the stratum corneum of the human skin and Vit E prevents lipid peroxidation in vivo [67]. The topical application of Vit E helps to reduce the length, width and frequency of fine lines and wrinkles [68].

In our study, the formulation with Vit E reduced the wrinkle size and vascular lesions by 19.8–27.5% and 21.35–69.05%, respectively, after 14 days of application. The discoloration changed by 12.93–44.89%, and in some measurements the level of melanin was detected as normal after the same period of time. In comparison, in the study by Hahn et al., a formulation containing Vit E reduced wrinkles by 14.07%. The skin tone and erythema (redness) also improved (showing a 22.39% reduction) [69]. In the case of the tested nanoemulsion with Vit E, the changes mentioned above were observed after 4 weeks of use (Figure 7).

## 4. Conclusions

Nanoemulsions are very promising systems for delivery of lipophilic compounds to the skin. Many properties related to their composition and physicochemical characteristics can influence their suitability and efficacy. A better understanding of these properties will help to extend the range of skin care product nanoformulations used in dermatology and cosmetics [50].

As part of this study, new SEDDS formulations containing SF as the surfactant, TR as the co-surfactant, encapsulated model oil Capmul and active substances (curcumin encapsulated in CA and ascorbyl tetraisopalmitate and tocopherol encapsulated in sunflower seed oil) were designed and created. Different compositions were prepared by varying the O/W ratio, the total surfactant content and the surfactant composition (expressed as an SF/TR ratio). The average particle sizes were low (69–183 nm), with similar particle size distribution behavior. The results of zeta potential assays indicated negative values in the range of −75.9 to −92.6 mV, which may partially explain the stability of the samples. The stability over time showed the emulsions to be stable after 195 days.

It was shown that the preparation remained in the skin layers. Hence, it can be supposed that its efficacy will be much higher than in the case of the classic application. Thanks to coumarin’s encapsulation in the nanoemulsion, it can penetrate the skin, stopping at the dermis border. The formulations with vitamin C and vitamin E reduced the discoloration, the vascular lesions and the depth of the wrinkles on the tested skin, which can be useful in cosmetics in the care for problem skin, including capillary and sensitive skin.

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
