# Peer review of "Nanoemulsion Stabilized by Safe Surfactin from Bacillus subtilis as a Multifunctional, Custom-Designed Smart Delivery System"

_pharmaceutics, 2020, doi:10.3390/pharmaceutics12100953_

Round 1

Reviewer 1 Report

This manuscript carries a very good idea of exploiting a bacterial-derived surfactant:surfactin in the development of nanoemulsions for cosmecutical purposes.

The idea is good and the design of experiments are fine. However, I have some concerns:

1- Please differentiate between the topical and transdermal delivery throughout the manuscript.

2- The manuscript seems to be dealing with a microemulsion system not a nano one. The authors should figure out the differences and distinguish between them. The method of preparation, the components and the obtained particle size suggests that the prepared system is a microemulsion not a nanoemulsion.

3- The different mechanisms of the microemulsion components penetration enhancement through the stratum corneum should be extensively discussed in the introduction.

4- The depth of skin penetration can be determined using the z-stack of the CLSM to give better comparisons.

5- The nature of the error bars in Figure 4 should be stated.

6- The caption of Figure 5 should be re-phrased.

Reviewer 2 Report

  • Consistency between Vit C/Vit E and vitamin C and E
  • Section 3.2.3 – No florescence images from control skin. Please add.
  • Lines 609 to 612 – add references where author names are mentioned.
  • Not convinced the studies described in section 2.6.4 and results discussed in section 3.2.4 are possible to conduct ex vivo
  • Reference 63 conducts this study in a clinical setting which is achievable. Ex vivo on freeze thawed skin is not the right way to do this study.
  • How was the skin maintain for 4 weeks? The description is lacking.
  • Reference 62 is not in English hence the reviewers can’t access the methods described.
  • This study compares all in vivo studies (66, 62, 63 etc.). Neither the comparison is valid or the methodology of Ex vivo studies. Suggest discuss the implications using literature but don’t include this study as part of the paper. The manuscript should be resubmitted after removing this experiment and tightening up the discussion surrounding.
  • Another alternative would be do conduct the study in vivo

Reviewer 3 Report

In this work, the authors prepared and characterized nanoemulsions using surfactin, a bio-surfactant obtained from Bacillus subtilis aiming at the delivery of vitamin E, C and curcumin in the deeper layers of the skin for the treatment of cutaneous damages including wrinkles, vascular lesions and discoloration.

The use of a bio-surfactant to produce nanoemulsions is an interesting approach but the experimental protocol is neither well designed nor well illustrated.

The introduction is too long and it should be shortened.

The statement at line 94 “A nanoemulsion can be used to deliver an active substance via transmucosal and transdermal routes, which is why such systems can effectively improve bioavailability” requires references. Please, add appropriate references.

The sentence at line 107 is a repetition of the sentence at line 54. Please, eliminate one of these sentences.

Line 265. The volume of the receptor chamber should be reported together with the surface area through which skin penetration occurred.

Line 268. The authors performed in vitro skin penetration studies for 1 h. Generally, in vitro skin penetration studies are performed for at least 6-8 h. Why did the authors use such a short time? After in vivo application of a pharmaceutical dosage form on the skin surface, the vehicle remains on the skin for several hours. What is the point of determining in vitro skin drug retention after 1 h?

Line 294. The authors reported, “The prepared 1 ml nanoformulations were applied to the skin. The measurements by means of the analyzer were carried out before the application of the formulations and 14 and 28 days after their application.” The procedure used to perform in vivo experiments is unclear. Generally, in vivo studies are performed using viscous vehicles to apply a defined amount of sample on a defined area of the skin. On the contrary, nanoemulsions have viscosity values similar to water. Being nanoemulsions liquid, how could the authors apply 1 ml of formulation on the skin surface? According to the description mentioned above, the authors applied only 1 ml of each formulation on the skin. What is the point of applying a single a dose of formulation evaluating the resulting effects after 14 and 28 days? How can such treatment provide an effect after 14 or even 28 days? What was the number of volunteers enrolled in the study? What was the age of these subjects? On which part of the body did the volunteers apply the formulations under investigation?

Line 423. In Table 3, the authors reported droplet size, polydispersity index and zeta potential of nanoemulsions containing CA, vitamin E, vitamin C and CAC. What is the percentage of the previously mentioned compounds in the nanoemulsions?

Line 591. As the authors aimed at delivering active ingredients into the skin, the term “transdermal” is incorrect. The term “transdermal” is used when the goal is to achieve a therapeutic level of drug into the systemic circulation. Please, correct.

Please, check the authors’ names in reference 25.

English should be revised.

Round 2

Reviewer 2 Report

Thank You for the clarification. I was sure that the work described can't be performed ex vivo.

Reviewer 3 Report

The authors addressed all comments.